# Potential demand for National Health Insurance in Zimbabwe: Evidence from selected urban informal sector clusters in Harare

Tamisai Chipunza[1]*, Senia Nhamo[2]

**1** Department of Economic Sciences, Midlands State University, Gweru, Zimbabwe, **2** Department of Economics, University of South Africa, Gauteng, South Africa

* tchipunzah@gmail.com

## Abstract

### Background

Zimbabwe's tax-based healthcare financing model has been characterised by perennial financing deficits and widespread application of user fees and has thus been socially exclusive. The country's urban informal sector population is not spared from these challenges. The study explored the potential demand for National Health Insurance (NHI) among respondents from selected urban informal sector clusters of Harare. The following clusters were targeted: Glenview furniture complex, Harare home industries, Mupedzanhamo flea market, Mbare new wholesale market and Mbare retail market.

### Methods

A cross-sectional survey was administered to 388 respondents from the selected clusters, and data on the determinants of Willingness to Join (WTJ) and Willingness to Pay (WTP) was gathered. Respondents were recruited via a multi-stage sampling procedure. In the first stage, the five informal sector clusters were purposely selected. The second stage involved a proportional allocation of respondents by cluster size. Finally, based on the stalls allocated by municipal authorities in each area, respondents were selected using systematic sampling. The sampling interval ($k$) was determined by dividing the total number of allocated stalls in a cluster ($N$) by the sample size proportionate to that cluster ($n$). For each cluster, the first stall (respondent) was randomly chosen, and thereafter, a respondent from every $10^{th}$ stall was selected and interviewed at their workplace. Contingent valuation was adopted to elicit WTP. Logit models and interval regression were applied for the econometric analyses.

### Results

A total of 388 respondents participated in the survey. The dominant informal sector activity among the surveyed clusters was the sale of clothing and shoes (39.2%), followed by the sale of agricultural products (27.1%). Concerning employment status, the majority were

**Data Availability Statement:** All relevant data are within the paper and its Supporting Information files

**Funding:** The author(s) received no specific funding for this work.

**Competing interests:** The authors have declared that no competing interests exist.

own-account workers (73.1%). Most of the respondents (84.8%) completed secondary school. On monthly income from informal sector activities, the highest frequency (37.1%) was observed in the Zw$(1000 to <3000) or US$(28.57 to <85.71) category. The mean age of respondents was 36 years. Out of the 388 respondents, 325 (83.8%) were willing to join the proposed NHI scheme. WTJ was influenced by the following factors: health insurance awareness, health insurance perception, membership to a resource-pooling scheme, solidarity with the sick, and household recently experiencing difficulties paying for healthcare. On average, respondents were willing to pay Zw$72.13 (approximately US$2.06) per person per month. The key determinants of WTP were household size, respondent's education level, income, and health insurance perception.

## Conclusions

Since the majority of respondents from the sampled clusters were willing to join and pay for the contributory NHI scheme, it follows that there is potential to implement the scheme for the urban informal sector workers from the clusters studied. However, some issues require careful consideration. The informal sector workers need to be educated on the concept of risk pooling and the benefits of being members of an NHI scheme. Household size and income are factors that require special attention when deciding on the premiums for the scheme. Moreover, given that price instability hurts financial products such as health insurance, there is a need for ensuring macroeconomic stability.

## Introduction

As a proportion of the official Gross Domestic Product (GDP), Zimbabwe's informal sector has grown from about 8% in 1980 to around 60% by 2018, making the country the world's second-biggest informal sector economy, after Bolivia [1,2]. In terms of employment, the informal sector accounts for more than 90% of Zimbabwe's labour force [3,4]. The rapid growth of the informal sector in Zimbabwe can be traced back to the early 1990s when the government implemented the Economic Structural Adjustment Programme (ESAP) which entailed, among other measures, a reduction in government expenditure and trade liberalisation. Deindustrialisation and massive retrenchment of workers following the implementation of ESAP between 1991 and 1995 led to the bourgeoning of Micro, Small and Medium Enterprises (MSMEs) [4–6].

The country's urban informal sector workers and their dependents face a high risk of impoverishment due to medical costs. Private health insurance in Zimbabwe caters for about 10% of the population, and membership in these schemes has been biased in favour of the formally employed [7]. The problem is compounded by limited fiscal allocations to the health sector which have generally stayed below the Abuja threshold of 15% of the national budget, leading to over-reliance on user fees [7–9]. Thus, Zimbabwe's health sector has generally been underfunded, with the problem becoming more pronounced during the crisis period (late 1990s to 2008). Public health expenditure fell to less than 1% of GDP [10], and per capita public health spending reached a record low level of US$0.02 [11] against a minimum of US$60 (2005 prices) per person required to attain the Millennium Development Goals (MDGs) [12]. Underfunding persisted even after the crisis period, causing the country to rely on Out of Pocket (OOP) payments and financing from development partners, each constituting about

25% of total health expenditure [7,8,13,14]. In fact, the Parliament of Zimbabwe [15] describes the country's current healthcare financing mechanism as an OOP model.

High levels of OOP health expenses cause social exclusion as some people are denied access to essential healthcare. This slows down a nation's progress towards Universal Health Coverage (UHC). Evidence from Costa Rica and Colombia show that insurance schemes designed in a pro-poor manner have a significant impact on access to healthcare, utilisation of health services, and financial protection for the population [16]. Ghana and Rwanda have also been cited as role models for the success of mandatory health insurance in moving towards UHC [17,18]. The Zimbabwean government planned to roll out National Health Insurance (NHI) by the year 2025 [19]. The implementation of an NHI scheme in the country can improve healthcare delivery as more resources for health become available. Moreover, membership in a viable NHI scheme entails reduced OOP health payments, hence a decrease in the likelihood of financial catastrophe due to sickness.

While several studies on Willingness to Pay (WTP) for improvements in the provision of public healthcare have been conducted in developed countries, similar studies are very scant in developing nations in Africa, Asia, and South America [20]. A literature search using the Google scholar engine, Scopus, Ebscohost, PubMed and ScienceDirect databases revealed that as of 18 February 2023, one Zimbabwean study [21] had applied the contingent valuation method to investigate the WTP for Community Based Health Insurance (CBHI) in Makoni district. The following search terms were used with no restrictions on publication date: "contingent valuation, health insurance, Zimbabwe". Bepe [21] applied Pearson's Chi-square test for the bivariate associations between WTP and some socio-demographic and health-related factors. The present study focuses on a contributory NHI scheme with reference to urban informal sector participants in Harare. Both bivariate analyses (to establish associations) and multivariate analyses (to test for causal relationships) are conducted. The results of the study are expected to add to the empirical literature on the potential demand for NHI among urban informal sector workers from a lower-middle-income country perspective.

The study explored the potential demand for a contributory NHI scheme among 388 urban informal sector participants from the following urban informal sector clusters in Harare: Glenview furniture complex, Harare home industries (formerly Mbare Siya So), Mupedzanhamo flea market, Mbare new wholesale market and Mbare retail market. A survey was administered to gather data on the determinants of Willingness to Join (WTJ) and WTP. A stated preference technique (contingent valuation) was adopted to elicit WTP. Logit regression models and an interval data model were applied for the regression analyses.

The rest of the paper is organised as follows: the next section presents the methodology of the study, followed by a presentation of the results. The results are then discussed, and thereafter the study limitations are highlighted. The final section gives the conclusions.

## Materials and methods

### Study design and setting

An exploratory cross-sectional research design was adopted. The following urban informal sector clusters in Harare were targeted: Glenview furniture complex, Harare home industries, Mupedzanhamo flea market, Mbare new wholesale market and Mbare retail market. These fall among the clusters with the highest concentration of informal sector activity in the country. Of these clusters, Glenview furniture complex and Harare home industries are characterised by some commendable degree of innovation in the manufacture of household furniture, steel products and agricultural implements.

## Population and sampling

Respondents were recruited via a multi-stage sampling procedure. In the first stage, the five informal sector clusters named under the study design and setting section were purposely selected. Municipal authorities allocated operating stalls to the informal sector workers in these clusters. The second stage involved a proportional allocation of respondents by cluster size. Finally, based on the stalls allocated by municipal authorities in each area, respondents were selected using systematic sampling. The sampling interval ($k$) was determined by dividing the total number of allocated stalls in a cluster ($N$) by the sample size proportionate to that cluster ($n$). For each cluster, the first stall (respondent) was randomly chosen, and thereafter, a respondent from every $10^{th}$ stall was selected and interviewed at their workplace. Purposively selecting the informal sector clusters in the first stage had some advantages. First, it helped to reduce survey costs since clusters with a high concentration of informal sector workers were targeted. Second, given that informal sector activities in Harare take place not only in designated places but also along certain streets in town and residential areas, targeting the informal sector clusters for which sampling frames were available from the City of Harare allowed for the application of probability sampling in selecting respondents. The sampling frame thus comprised informal sector workers granted permission to operate and allocated a stall from which they do business.

According to statistics from the City of Harare's Human Capital Department, the total number of participants allocated stalls in the targeted informal sector clusters (which constituted the population of this study) was 4384. The population was distributed as follows: Glenview furniture complex (483), Harare home industries (911), Mupedzanhamo flea market (1442), Mbare new wholesale market (153) and Mbare retail market (1395). Based on Raosoft [22], the minimum sample size to achieve a 5% margin of error, 95% confidence interval, and 50% response distribution from the target population was 354. The researcher allowed for a 15% non-response rate and thus targeted a total of 408 respondents. This number was proportionately allocated among the five clusters as follows: Glenview furniture complex (45), Harare home industries (85), Mupedzanhamo flea market (134), Mbare new wholesale market (14) and Mbare retail market (130).

## Data collection

Data collection took place between February and March 2020. The survey mode was a structured (closed-ended) questionnaire that was interviewer-administered. Before the main survey, a pilot test was conducted with 30 respondents from Harare home industries, one of the clusters for the main study.

## Eliciting willingness to join and pay

The healthcare package and initial bids for this contingent valuation study were determined with reference to private medical care packages that were being offered by the country's seasoned private health insurance companies as of January 2020 when a pilot survey for the study was conducted. The researcher established that the private health insurance companies were charging monthly premiums per person which were, on average, 0.25% of the annual global/family benefit limit. Based on these benefit limits and the market exchange rate of US$1: Zw $35 for that period [23], the benefit limit for the contributory NHI scheme was set at Zw $40000 (about US$1143) per family per year. Applying the 0.25% rule gave a monthly contribution of Zw$100 per person (approximately US$2.86). The contributory NHI scheme assumed that the government and citizens had a joint responsibility in financing healthcare,

and thus involved member contributions and a government subsidy. It was presented to respondents as follows:

"Suppose government introduces a subsidised health insurance scheme for the informal sector whereby those who join and contribute some specified monthly premiums will not pay for hospitalisation and specialist treatment at public and private health facilities, consultation at medical practitioners, maternity, blood transfusion, dental care, laboratory tests, x-ray and scan services, and prescription drugs up to a family limit of Zw$40000 (**i.e., BONDS**) per year."

The starting premiums in the bidding game were determined by assuming government subsidies of 75%, 50%, 25%, and zero. Given these possible levels of government subsidies and the average private-sector premium of Zw$100 computed above, the initial bids took values of Zw$25, Zw$50, Zw$75, and Zw$100, respectively. The researcher randomly offered one of these initial bids to a respondent and this was done to alleviate the problem of bid anchoring [24,25]. This first round of elicitation was followed by a second round whereby a higher or lower bid was offered depending on the answer to the initial offer. If the respondent indicated WTP the first premium, a higher bid was offered as a follow-up. Conversely, if the respondent rejected the initial bid, a lower bid was offered. Thus, WTP for NHI was elicited using the Double-Bounded Dichotomous Choice (DBDC) with follow-up method. After the DBDC with follow-up elicitation, an open-ended question on the respondent's maximum WTP was asked, and contingent valuation studies that have included this question exist [26,27].

It should be noted that besides the DBDC with follow-up format, researchers have elicited WTP using open-ended questions, payment cards, bidding games, and the Single-Bounded Dichotomous Choice (SBDC) method. When compared to the DBDC with follow-up method, the alternative elicitation formats have limitations such as a high rate of non-response, high rate of zero WTP values, range bias, and inefficient WTP estimates [28–30]. Due to these limitations, some elicitation formats have become less popular–generally there has been growing use of bidding games and dichotomous-choice formats in comparison to open-ended questions and payment cards [31]. Concerning the dichotomous-choice formats, whilst the SBDC format is more incentive compatible (encourages truth-telling by respondents) [29,32], it is statistically inefficient (has lower precision in determining WTP) when compared to the DBDC format [33].

## Measures to enhance validity and reliability

Various measures were taken to enhance the credibility (validity and reliability) of the survey research results. The selection of a large, representative sample through probability sampling helped in reducing non-coverage and sampling bias. To ensure that the survey questions were well-understood by respondents (minimising measurement bias), survey pre-testing was done with 30 respondents drawn from one of the clusters for the main survey. Moreover, the survey was interviewer-administered and this had the advantage of obtaining a relatively high response rate, thus minimising non-response bias. In addition, most of the questions required a recall period of at most three months and this helped to minimise recall bias. Also, in line with the University of South Africa (UNISA) research ethics requirements, the questionnaire did not contain questions that required respondents to disclose special personal information such as health status, religion, or political persuasion.

To improve the incentive compatibility of the elicitation formats, respondents were presented with a coercive payment vehicle in the form of insurance premiums; a payment vehicle that sounds realistic for the commodity being valued (NHI), and at the same time ruling out the possibility of free riding. Moreover, as part of the introductions, the researcher emphasised

having been cleared by the Ministry of Health and Child Care (MoHCC) as well as the Department of Economics Ethics Review Committee at UNISA. This helped in creating confidence among respondents, and they likely perceived their responses as being consequential (that is, being inputs to a change in Zimbabwe's healthcare financing policy). This strategy encourages incentive compatibility [34,35].

As much as respondents were informed of the research having been approved by the MoHCC and thus being consequential, the researcher emphasised that respondents should feel free to give their honest responses which were kept confidential. Further emphasis was put on participation being voluntary and that the respondent could withdraw any time they wished to do so. Moreover, during every round of data collection, the researcher wore the UNISA student identity card around their neck to establish their credibility and allay fears of the research being done for covert purposes. All these measures were meant to guard against the risk of getting coercive responses; in some settings, fear of government authorities may influence respondents' attitudes [29].

## Ethical consideration

The research obtained ethics clearance from the Department of Economics Ethics Review Committee, UNISA (reference number: 2019_DE_18(PD)_T CHIPUNZA). Before applying for ethics clearance at UNISA, permission to conduct the research had been granted by MoHCC, Zimbabwe. Moreover, participation was voluntary and respondents signed the informed consent form before responding to the questionnaire.

## Modelling Willingness to Join and Willingness to Pay

The theoretical foundation for modelling WTP is the random utility theory [36] and Lancaster [37]'s characteristics model [38–41]. The characteristics model posits that the consumer derives satisfaction from the attributes of a commodity rather than merely from the product. According to the random utility theory and the characteristics model, it is assumed that individuals aim to maximise utility, and thus a proposed intervention with the greatest satisfaction is chosen from the available alternatives. Various factors influence the respondent's choice of an improved alternative, and these include the other offers available, the respondent's socio-economic characteristics (for example, income, age, and education), and the bids on offer [39].

**Logit model for WTJ.** When the regressand is binary (for example, "yes" or "no" responses), estimation is performed using the logit and probit models. Given that the regression results from logit and probit models are generally similar, researchers normally choose one of the models depending on model adequacy as portrayed by the Hosmer-Lemeshow test. In this study, the goodness of fit of the two models was generally the same (p > 0.84 in both WTJ and WTP regressions). The logit model was adopted as it also allowed for the reporting and interpretation of odds ratios. Concerning WTJ, the regressand is binary and takes a value of one when a respondent is willing to join and zero when they are not willing to join. Thus, based on the logit model, the probability of WTJ the NHI scheme is mathematically represented by Eq 1:

$$p_i = \frac{e^{z_i}}{1 + e^{z_i}} \qquad (1)$$

where $p_i$ is the probability that a respondent is willing to join.

$z_i$ is a function of $n$ regressors and is expressed as follows:

$$Z_i = \beta_0 + \beta_1 X_1 + \beta_2 X_2 + \cdots + \beta_n X_n \qquad (2)$$

where $\beta_0$ is the intercept, $\beta_1 \ldots \beta_n$ are the regression coefficients, and $X_1 \ldots X_n$ are the explanatory variables.

**Logit model for WTP.**   Once again, in this SBDC elicitation, WTP is a binary variable taking a value of one if $WTP>0$, and zero if $WTP=0$.

The model is presented as follows:

$$Y_i^* = \hat{X}\beta + \varepsilon_i \tag{3}$$

where $Y_i^*$ is an unobservable latent variable (in this case WTP); X is a vector of explanatory variables; $\beta$ is a vector of regression coefficients; and $\varepsilon$ is an error term that follows the logistic distribution.

**Interval regression model for WTP.**   In the WTP literature, some studies modelled WTP data from the DBDC elicitation via the Ordinary Least Squares (OLS) technique [27,42,43] whilst others embraced the maximum likelihood method [24,26,40,41,44,45]. It is argued that when the regressand is bounded between two points (meaning that there is no distinct point estimate), estimation should be done via maximum likelihood rather than OLS [40,44]. However, in cases where respondents also state the maximum WTP and the dependent variable is not censored at zero, the regressand will be a continuous variable with a single point estimate such that the OLS technique can be applied [27].

In this study, WTP was estimated using a maximum likelihood approach—the interval data model based on Lopez-Feldman's 'doubleb' command [24,26,27,46]. From the DBDC elicitation, four outcomes are possible, each with a certain probability of occurrence, that is, Pr.(Yes, Yes), Pr.(Yes, No), Pr.(No, Yes), and Pr.(No, No). Following Florio and Giffoni [26], the indicator functions associated with these response paths are represented by $I_i^{YY}$, $I_i^{YN}$, $I_i^{NY}$, and $I_i^{NN}$, respectively. In this case, the value of $I(.)$ is zero or one depending on each respondent's case. Thus, the function to be maximised is:

$$lnL(\theta) = \sum_{i-1}^{N} \{I_i^{YY}\ln[1 - G(t_i^u; \theta)] + I_i^{YN}\ln[G(t_i^u; \theta) - G(t_i^u; \theta)] + I_i^{NY}\ln[G(t_i^0; \theta) - G(t_i^l; \theta)] + I_i^{NN}\ln G(t_i^l; \theta)\} \tag{4}$$

As shown in Eq (4), the indicator functions, $I(.)$, imply that the contribution of each respondent to the likelihood function is just one out of four possibilities.

The econometric model for WTP is expressed as:

$$WTP_i(X_i, \varepsilon_i) = X_i\beta + \varepsilon_i \tag{5}$$

where $X_i$ is a vector of explanatory variables; $\beta$ is a vector of regression coefficients; and $\varepsilon_i$ is an error term with mean zero and variance $\delta^2$.

## Description of independent variables

Table 1 gives a description of the variables used in the univariate, bivariate, and multivariate analyses. The expected relationships with WTJ and WTP are also given.

## Results

Data analysis for WTJ and WTP was performed using Intercooled Stata (version 13). It is classified into three broad sections, namely univariate descriptive analyses, bivariate analyses, and multivariate analyses.

### Response rate

From the targeted 408 respondents, 388 participated in the study. Thus, the response rate was 95%.

**Table 1. Description of independent variables and a priori relationship with willingness to join and pay for NHI.**

| Variable | Explanation | Measurement | A priori relationship with willingness to join and pay |
|---|---|---|---|
| Gender | Whether respondent is male or female | 0 = Female<br>1 = Male | Being male is associated with more WTJ and WTP since men are more involved in deciding on the family budget. |
| Age | Respondent's age | Continuous (years) | WTJ and WTP increases with age. |
| Age squared | Square of respondent's age | Continuous (years) | In old age, individuals become less willing to join and pay. |
| Marital status | Respondent's marital status | 0 = Unmarried (single/ divorced/ widowed)<br>1 = Married | It is difficult to predict the relationship between marital status and WTJ and WTP. |
| Education | Highest level of education completed by respondent | 1 = No education<br>2 = Primary<br>3 = Secondary<br>4 = Tertiary | Individuals that are more educated value the importance of medical insurance and have higher WTJ and WTP. |
| Household size | Number of members in a household | Continuous (number) | The larger the household size the greater the WTJ and WTP. |
| Income | Average monthly income of the household from informal sector activities | 1 = < Zw$500<br>2 = Zw$500 to < Zw$1000<br>3 = Zw$1000 to < Zw$3000<br>4 = ≥ Zw$3000 | The higher the household income the higher the WTJ and WTP. |
| Distance | Distance from respondent's residence to the nearest health facility | Kilometres | The longer the distance to the nearest health facility the less the WTJ and WTP. |
| Membership | Whether the respondent is a member of a resource pooling scheme such as a savings & lending club, health insurance or funeral fund | 0 = No<br>1 = Yes | Being a member of a resource-pooling scheme positively influences WTJ and WTP. |
| Awareness | Whether the respondent is aware of private health insurance schemes in the country | 0 = No<br>1 = Yes | Awareness of health insurance positively impacts WTJ and WTP. |
| Perception | Respondent's perception on whether health insurance protects members from high medical costs | 0 = No<br>1 = Yes<br>2 = Not sure | Respondents with a positive view of the benefits of health insurance in protecting patients from high medical costs are more willing to join and pay |
| Solidarity | Whether the respondent supports the use of their money to pay healthcare expenses for someone sicker than them | 0 = No<br>1 = Yes | WTP health expenses for the sick reflects society's level of solidarity with the sick and hence higher WTJ and WTP for NHI. |
| Difficulties | Whether the household experienced difficulties paying for healthcare during the past 12 months | 0 = No<br>1 = Yes | Respondents whose household experienced difficulties paying for healthcare in the past are more willing to join and pay. |

## Univariate descriptive analyses

As shown in Table 2, the dominant informal sector activity among the surveyed clusters was the sale of clothing and shoes (39.2%), followed by the sale of agricultural products (27.1%) and the timber and furniture business (15%). With regard to gender, most respondents were male (71.4%).

Fig 1A–1C show the distributions of education, income and employment status. The majority of respondents (84.8%) completed secondary school. On monthly income from informal sector activities, the highest frequencies (29.9% and 37.1%) were observed in the Zw$(500 to <1000) and the Zw$(1000 to <3000) categories, respectively. Considering the market exchange rate that prevailed during the survey period (around US$1: Zw$35) [23], the income categories with the highest frequencies in the study might seem low. However, those categories were informed by the results of a pilot test and they compared very well with the new salaries that had just been offered to professionals working in the country's public service [47]. Concerning employment status, the majority were own-account workers (73.1%)—contributing family members constituted 10.6% of the respondents, 14.5% were employees, and 1.8% were employers.

**Table 2. Respondents' demographic and socio-economic profiles.**

| Variable | Category | Frequency | Percentage |
|---|---|---|---|
| Informal sector activity | Clothing & shoe sales | 152 | 38.2 |
| | Hardware | 25 | 6.4 |
| | Food kiosk | 11 | 2.8 |
| | Repairs | 2 | 0.5 |
| | Metal business | 35 | 9.0 |
| | Timber & furniture business | 58 | 15.0 |
| | Agricultural products | 105 | 27.1 |
| Respondent's employment status | Own-account worker | 283 | 73.1 |
| | Employee | 56 | 14.5 |
| | Employer | 7 | 1.8 |
| | Contributing family member | 41 | 10.6 |
| Gender | Female | 111 | 28.6 |
| | Male | 277 | 71.4 |
| Marst | Unmarried (single/ divorced/ widowed) | 82 | 21.1 |
| | Married | 306 | 78.9 |
| Education | No education | - | - |
| | Primary | 30 | 7.7 |
| | Secondary | 329 | 84.8 |
| | Tertiary | 29 | 7.5 |
| Income | < Zw$500 | 34 | 8.8 |
| | Zw$500 to < Zw$1000 | 116 | 29.9 |
| | Zw$1000 to < Zw$3000 | 144 | 37.1 |
| | ≥ Zw$3000 | 94 | 24.2 |

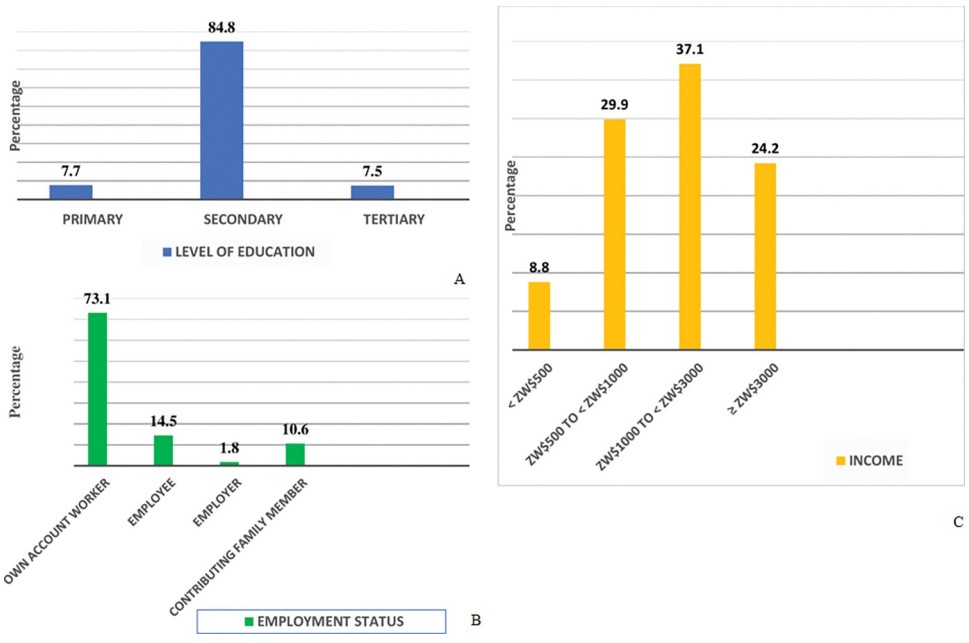

**Fig 1. Distribution of education, income and employment.**

**Table 3. Healthcare- and health insurance-related issues.**

| Variable | Category | Frequency | Percentage |
|---|---|---|---|
| **Healthcare-related issues** | | | |
| Health expenditure in the past 3 months | None | 125 | 32.2 |
| | < Zw$100 | 62 | 16.0 |
| | Zw$100 to < Zw$500 | 115 | 29.6 |
| | Zw$500 to < Zw$1000 | 55 | 14.2 |
| | ≥ Zw$1000 | 31 | 8.0 |
| Difficulties | No | 98 | 25.3 |
| | Yes | 290 | 74.7 |
| Respondent's perception of healthcare costs | Affordable | 14 | 3.6 |
| | Expensive | 80 | 20.7 |
| | Very expensive | 293 | 75.7 |
| Quality of public healthcare | Poor | 332 | 85.6 |
| | Satisfactory | 27 | 7.0 |
| | Good | 26 | 6.7 |
| | Very good | 3 | 0.8 |
| **Health insurance-related issues** | | | |
| Awareness | No | 59 | 15.2 |
| | Yes | 329 | 84.8 |
| Health insurance ownership | No | 343 | 88.4 |
| | Yes | 45 | 11.6 |
| Perception | No | 114 | 29.5 |
| | Yes | 231 | 59.7 |
| | Not sure | 42 | 10.8 |
| Membership | No | 131 | 33.9 |
| | Yes | 256 | 66.1 |
| Solidarity | No | 40 | 10.3 |
| | Yes | 348 | 89.7 |
| Preferred healthcare financing mechanism | National Health Insurance | 252 | 65.0 |
| | Direct payments | 22 | 5.7 |
| | Private health insurance | 33 | 8.5 |
| | General Government taxes | 81 | 20.9 |

The frequency distributions of health and healthcare-related information are shown in Table 3. The table shows that most respondents (67.8%) reported having incurred direct or OOP medical expenses during the last three months. 74.7% of the respondents indicated having faced difficulties paying for healthcare in the past 12 months, and 75.7% perceived healthcare in Zimbabwe as very expensive. 85.6% of the respondents described the country's public healthcare delivery as poor. Concerning membership to a resource-pooling scheme such as a funeral fund, private health insurance scheme or savings and lending club, 66.1% of respondents reported that they were members, and on the best healthcare financing mechanism for the country, 65% were in favour of NHI. 89.7% of the respondents were in favour of financially supporting the sick, indicating a high level of solidarity on health matters.

The summary statistics for continuous variables in Table 4 show that the age of respondents ranged between 18 and 62 years, and the mean age was 36 years. Household size ranged from 1 to 13 members, with a mean household size of 4.6. On average, the distance from the respondent's place of residence to the nearest health facility was 2.45 kilometres. The Kernel density plots (Fig 2A–2C) show the distribution of age, household size and distance. Table 4 also

**Table 4. Summary statistics.**

| Variable | Observations | Mean | Standard deviation | Minimum | Maximum |
|---|---|---|---|---|---|
| Age (years) | 388 | 36.22 | 9.15 | 18 | 62 |
| Household size (number) | 388 | 4.58 | 1.75 | 1 | 13 |
| Period involved in informal sector (years) | 388 | 8.58 | 6.11 | 1 month | 32 |
| Number of workers at an establishment | 387 | 2.23 | 1.36 | 1 | 9 |
| Distance (kilometres) | 388 | 2.45 | 1.95 | 0.1 | 11 |

shows that some respondents were engaged in the informal sector for only one month (0.083 years) while others had been in the sector for 32 years. On average, an establishment had 2 workers (inclusive of the owner). With regards to overall WTJ, out of the 388 respondents, 325 (83.8%) were willing to join the NHI scheme. Of the 325 indicated willingness to join, 208 (64%) answered "yes" to paying the first bid.

## Bivariate analyses

Tables 5 and 6 summarise the results of hypothesis tests for the association between selected variables and WTJ, as well as WTP the first bid. The former reports the results on categorical covariates whilst the later shows the results for continuous covariates.

On the issue of WTJ, significant associations were established with the following factors: difficulties paying for healthcare, preferred healthcare financing mechanism, membership to a resource-pooling scheme, health insurance awareness, health insurance perception, and desire to support others in times of sickness. The Chi-square test was significant ($p < 0.05$) for these factors. Concerning hypothesis tests for WTP the first bid and the categorical variables, three factors (marital status, education, and income) were significant. As reported in Table 6, the Wilcoxon rank-sum test established that household size and distance to the nearest health facility were significantly associated with WTP the first bid at the 5% level ($p < 0.05$).

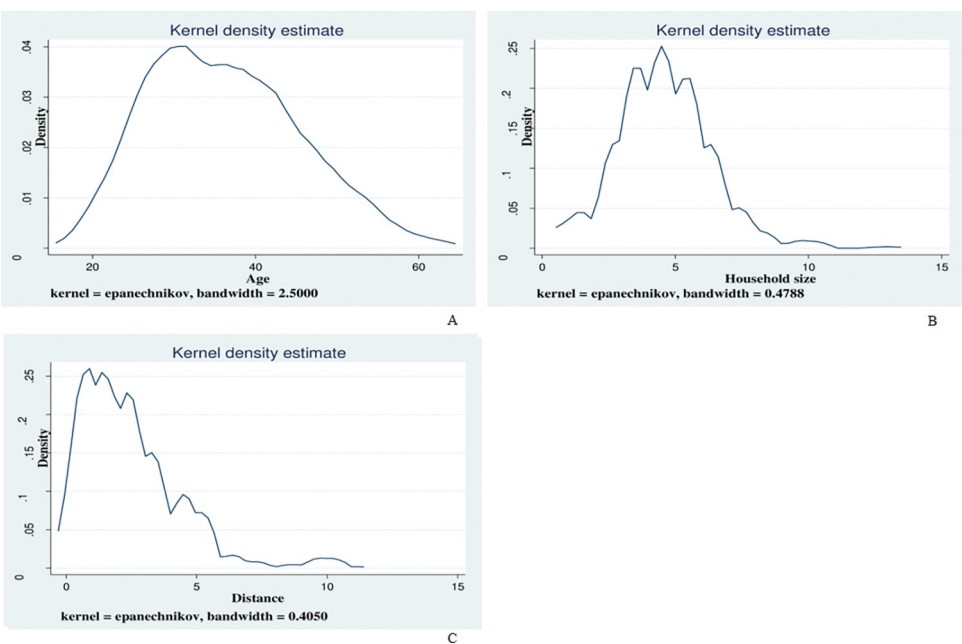

**Fig 2. Kernel densities of age, household size and distance.**

**Table 5. Frequency distribution and Chi-square analysis of willingness to join and pay the first bid (categorical covariates).**

| Variable | Willingness to join | | | | Willingness to pay the first bid | | | |
|---|---|---|---|---|---|---|---|---|
| | Freq | Yes | No | p-value | Freq | Yes | No | p-value |
| Gender | | | | | | | | |
| Female | 111 | 93 (83.8%) | 18 (16.2%) | 0.994 | 93 | 55 (59.1%) | 38 (40.9%) | 0.248 |
| Male | 277 | 232 (83.7%) | 45 (16.3%) | | 232 | 153 (66%) | 79 (34%) | |
| Marital status | | | | | | | | |
| Unmarried | 82 | 67 (81.7%) | 15 (18.3%) | 0.570 | 67 | 51 (76.1%) | 16 (23.9%) | 0.020 ** |
| Married | 306 | 258 (84.3%) | 48 (15.7%) | | 258 | 157 (60.9%) | 101 (39.1%) | |
| Education | | | | | | | | |
| No education | - | - | - | 0.396 | - | - | - | 0.017 ** |
| Primary | 30 | 23 (76.7%) | 7 (23.3%) | | 23 | 9 (39.1%) | 14 (60.9% | |
| Secondary | 329 | 276 (83.9%) | 53 (16.1%) | | 276 | 179 (64.9%) | 97 (35.1%) | |
| Tertiary | 29 | 26 (89.7%) | 3 (10.3%) | | 26 | 20 (76.9%) | 6 (23.1%) | |
| Income | | | | | | | | |
| < Zw$500 | 34 | 25 (73.5%) | 9 (26.5%) | 0.374 | 25 | 13 (52%) | 12 (48%) | 0.001 *** |
| Zw$500 to < Zw$1000 | 116 | 97 (83.6%) | 19 (16.4%) | | 97 | 53 (54.6%) | 44 (45.4%) | |
| Zw$1000 to < Zw$3000 | 144 | 122 (84.7%) | 22 (15.3%) | | 122 | 76 (62.3%) | 46 (37.7%) | |
| ≥ Zw$3000 | 94 | 81 (86.2%) | 13 (13.8%) | | 81 | 66 (81.5%) | 15 (18.5%) | |
| Difficulties | | | | | | | | |
| No | 98 | 63 (64.3%) | 35 (35.7%) | <0.001 *** | 63 | 39 (61.9%) | 24 (38.1%) | 0.700 |
| Yes | 290 | 262 (90.3%) | 28 (9.7%) | | 262 | 169 (64.5%) | 93 (35.5%) | |
| Preferred healthcare financing mechanism | | | | | | | | |
| National Health Insurance | 252 | 252 (100%) | - | <0.001 *** | 252 | 158 (62.7%) | 94 (37.3%) | 0.214 |
| Direct payments | 22 | 11 (50%) | 11 (50%) | | 11 | 6 (54.5%) | 5 (45.5%) | |
| Private health insurance | 33 | 16 (48.5%) | 17 (51.5%) | | 16 | 14 (87.5%) | 2 (12.5%) | |
| General Government taxes | 81 | 46 (56.8%) | 35 (43.2%) | | 46 | 30 (65.2%) | 16 (34.8%) | |
| Membership | | | | | | | | |
| No | 131 | 94 (71.8%) | 37 (28.2%) | <0.001 *** | 94 | 69 (68.8%) | 35 (37.2%) | 0.788 |
| Yes | 256 | 230 (89.8%) | 26 (10.2%) | | 230 | 148 (64.4%) | 82 (35.6%) | |
| Awareness | | | | | | | | |
| No | 59 | 32 (54.2%) | 27 (45.8%) | <0.001 *** | 32 | 22 (68.7) | 10 (31.3) | 0.555 |
| Yes | 329 | 293 (89.1) | 36 (10.9) | | 293 | 186 (63.5) | 107 (36.5) | |

*(Continued)*

**Table 5.** (Continued)

| Variable | Willingness to join | | | | Willingness to pay the first bid | | | |
|---|---|---|---|---|---|---|---|---|
| | Freq | Yes | No | p-value | Freq | Yes | No | p-value |
| Perception | | | | | | | | |
| No | 114 | 85 (74.6%) | 29 (25.4%) | <0.001 *** | 85 | 46 (54.1%) | 39 (45.9%) | 0.062 |
| Yes | 231 | 209 (90.4%) | 22 (9.5%) | | 209 | 140 (67.0%) | 69 (33.0%) | |
| Not sure | 42 | 30 (71.4%) | 12 (28.6%) | | 30 | 22 (73.3%) | 8 (26.7%) | |
| Solidarity | | | | | | | | |
| No | 40 | 17 (42.5%) | 23 (57.5%) | <0.001 *** | 17 | 8 (47.1%) | 9 (52.9%) | 0.135 |
| Yes | 348 | 308 (88.5%) | 40 (11.5%) | | 308 | 200 (64.9%) | 108 (35.1%) | |
| Quality of public healthcare | | | | | | | | |
| Poor | 332 | 279 (84.0%) | 53 (16.0%) | 0.616 | 279 | 176 (63.1%) | 103 (36.9%) | 0.221 |
| Satisfactory | 27 | 21 (77.8)% | 6 (22.2%) | | 21 | 17 (81%) | 4 (19.0%) | |
| Good | 26 | 23 (88.5%) | 3 (11.5%) | | 23 | 13 (56.5%) | 10 (43.5%) | |
| Very good | 3 | 2 (66.7%) | 1 (33.3%) | | 2 | 2 (100%) | 0 (0%) | |

Significance levels are shown by asterisks (

***$p < 0.01$

**$p < 0.05$).

## Multivariate analyses

**Willingness to join the proposed NHI scheme.** Logit regression results on WTJ the NHI scheme, together with the marginal effects, are presented in Table 7. As shown in Table 7, the Chi-square test for the log-likelihood ratio is statistically significant ($p < 0.0001$). It follows that the model is significant. From the logit regression results, the following factors increase the likelihood that someone is willing to join NHI: being a member of a resource-pooling scheme, health insurance awareness, solidarity with the sick, and household having faced difficulties paying for healthcare in the last 12 months. The marginal effect of membership shows that all else remaining constant, being a member of a resource-pooling scheme (health insurance, savings and lending club, funeral fund) increases the likelihood of WTJ the proposed

**Table 6. Wilcoxon rank-sum tests of willingness to join and pay the first bid (continuous covariates).**

| Variable | Willingness to join | | Willingness to pay the first bid | |
|---|---|---|---|---|
| | Z score | p-value | Z score | p-value |
| Age | 0.756 | 0.4497 | 1.750 | 0.0801 |
| Household size | 1.194 | 0.2326 | 2.610 | 0.0091 *** |
| Distance | 1.057 | 0.2904 | 2.325 | 0.0201 ** |

Significance levels are shown by asterisks (

***$p < 0.01$

**$p < 0.05$).

**Table 7. Logit regression results of Willingness to Join and the marginal effects.**

| Variable | Coefficient | Z statistic | P-value | Marginal effects |
|---|---|---|---|---|
| Age | -0.0200 (0.0223) | -0.90 | 0.369 | -0.0017 |
| Gender (ref: female) Male | 0.1400 (0.4093) | 0.34 | 0.732 | 0.1190 |
| Marital status (ref: unmarried) Married | -0.1913 (0.5149) | -0.37 | 0.710 | 0.0156 |
| Household size | -0.9552 (0.1084) | -.088 | 0.378 | -0.0080 |
| Membership (ref: no) Yes | 1.1444 (0.3868) | 2.96 | 0.003*** | 0.1065 |
| Awareness (ref: no) Yes | 2.8256 (0.5586) | 5.06 | 0.000*** | 0.3696 |
| Perception (ref: no) Yes Not sure | 0.5600 (0.4148) 1.4754 (0.6713) | 1.35 2.20 | 0.177 0.028** | 0.0510 0.1133 |
| Education (ref: primary) Secondary Tertiary | 0.4300 (0.6341) 0.5412 (0.9743) | 0.68 0.56 | 0.498 0.579 | 0.0391 0.0481 |
| Income (ref: < Zw$500) Zw$500 to < Zw$1000 Zw$1000 to < Zw$3000 ≥ Zw$3000 | 0.8132 (0.6401) 0.6434 (0.6146) 0.7631 (0.6712) | 1.27 1.05 1.14 | 0.204 0.295 0.256 | 0.0761 0.0622 0.0721 |
| Distance | 0.0679 (0.0932) | 0.73 | 0.467 | 0.0057 |
| Solidarity (ref: no) Yes | 2.4413 (0.4693) | 5.20 | 0.000*** | 0.3196 |
| Difficulties (ref: no) Yes | 1.6608 (0.3775) | 4.40 | 0.000*** | 0.1712 |
| Quality (ref: poor) Satisfactory Good Very good | 0.2214 (0.6571) 0.8282 (0.9135) 0.2310 (1.8955) | 0.34 0.91 0.12 | 0.736 0.365 0.903 | 0.0182 0.0600 0.0190 |
| Constant: Log likelihood Prob > Chi² Pseudo R² Observations | -4.7966 -109.4714 0.0000 0.3626 386 | | | |

Standard errors are shown in parentheses.

Significance levels are shown by asterisks (

***p < 0.01

**p < 0.05).

Marginal effects were computed as average marginal effects.

NHI scheme by 10.65% on average compared to not being a member (omitted category). Moreover, being aware of the activities of private health insurance schemes in Zimbabwe increases the probability of WTJ the proposed NHI on average by 36.96% than being unaware (reference category), ceteris paribus.

On the solidarity variable, ceteris paribus, respondents who consented to paying healthcare expenses for others sicker than themselves were, on average, 31.96% more likely to be willing to join than those who were not in favour of the solidarity principle (base category). Lastly, other things being equal, respondents whose families had experienced difficulties paying for healthcare had a probability of 17.12%, on average, of willing to join the NHI scheme than those that had not faced difficulties (reference category).

In addition to the logit regression results presented above, the odds ratios were also estimated and the results are shown in Table 8. The odds ratio of 3.1406 for "membership" shows that respondents who were current members of a resource-pooling scheme were 3.1406 times

**Table 8. Odds ratios for Willingness to Join.**

| Variable | Odds ratio | Standard error | z-statistic | p-value |
|---|---|---|---|---|
| Age | 0.9802 | -0.90 | -0.90 | 0.369 |
| Gender (ref: female)<br>Male | 1.1502 | 0.4708 | 0.34 | 0.732 |
| Marital status (ref: unmarried)<br>Married | 0.8259 | 0.4253 | -0.37 | 0.710 |
| Household size | 0.9089 | 0.0985 | -0.88 | 0.378 |
| Membership (ref: no)<br>Yes | 3.1406 | 1.2148 | 2.96 | 0.003*** |
| Awareness (ref: no)<br>Yes | 16.8715 | 9.4236 | 5.20 | 0.000*** |
| Perception (ref: no)<br>Yes<br>Not sure | 1.7506<br>4.3728 | 0.7262<br>2.9353 | 1.35<br>2.20 | 0.177<br>0.028** |
| Education (ref: primary)<br>Secondary<br>Tertiary | 1.5373<br>1.7181 | 0.9748<br>1.6739 | 0.68<br>0.56 | 0.498<br>0.579 |
| Income (ref: < Zw$500)<br>Zw$500 to < Zw$1000<br>Zw$1000 to < Zw$3000<br>≥ Zw$3000 | 2.2550<br>1.9029<br>2.1449 | 1.4435<br>1.1695<br>1.4396 | 1.27<br>1.05<br>1.14 | 0.204<br>0.295<br>0.256 |
| Distance | 1.0701 | 0.0998 | 0.73 | 0.467 |
| Solidarity (ref: no)<br>Yes | 11.4876 | 5.3906 | 5.20 | 0.000*** |
| Difficulties (ref: no)<br>Yes | 5.2635 | 1.9868 | 4.40 | 0.000*** |
| Quality (ref: poor)<br>Satisfactory<br>Good<br>Very good | 1.2479<br>2.2893<br>1.2598 | 0.8200<br>2.0913<br>2.3880 | 0.34<br>0.91<br>0.12 | 0.736<br>0.365<br>0.903 |

Constant: 0.0083
Log likelihood: -109.4714
Prob > Chi$^2$: < 0.0001
Pseudo R$^2$: 0.3626
Observations: 386

Significance levels are shown by asterisks (

***p < 0.01

**p < 0.05).

more likely to be willing to join the NHI scheme. Similarly, respondents who had health insurance awareness were 16.8715 times more likely to answer yes to the WTJ question. With respect to solidarity and having experienced difficulties paying for healthcare, those in favour of financially supporting the sick were 11.4876 times more likely to be willing to join while those who faced challenges meeting treatment costs in the last 12 months were 5.263 times more likely to be willing to participate in the proposed NHI scheme.

**Willingness to pay for NHI.** *Results of the Single-Bounded Dichotomous Choice (SBDC) elicitation.* Regression results of the logit model (Table 9) show that the factors that influenced WTP for NHI among the sampled informal sector clusters were household size, health insurance perception, education, income, and the bid price (initial bid offered to a respondent). Except for household size and the initial bid price (with negative significant coefficients), all the other factors significantly positively influenced WTP.

The marginal effect of household size shows that ceteris paribus, for each additional household member the likelihood of WTP for NHI decreases by 3.64%, on average. Moreover, respondents with a positive perception of health insurance have a 12.71% probability, on average, of willing to pay compared to those with a negative perception (omitted category). Other conditions held constant, having secondary and tertiary education increases the likelihood of WTP for NHI by 22.58% and 31.86% (on average), respectively in comparison to having primary education (reference category). Moreover, all else remaining the same, being in the high-income category ($\geq$ Zw\$3000 per month) than in the lowest income category ($<$ Zw\$500 each month) increases the probability of WTP by 19.7%, on average. In addition, the bid price variable was negatively related to WTP and was significant (p $<$ 0.01).

The odds ratios following the logit results presented above are reported in Table 10. A significant variable with an odds ratio greater than unity (one) implies an increased likelihood to pay and the reverse is true for odds ratios less than one. Concerning household size, the odds ratio of 0.7750 suggests that for each additional household member, WTP reduced 0.225 times. Respondents with a positive perception of health insurance were 2.3465 times more likely to be willing to pay, while those in the high-income category ($\geq$ Zw\$3000) were 4.1981 times more likely to be willing to pay. With regards to education, those with secondary and tertiary education were 4.35 times and 8.6601 times more willing to pay, respectively.

*Results of the Double-Bounded Dichotomous Choice (DBDC) with follow-up elicitation.* With respect to the determinants of WTP based on the DBDC with follow-up elicitation, household size, health insurance perception, education and income significantly influenced WTP. Table 11 summarises the interval regression results. The results show that household size was negatively related to WTP, implying that WTP decreased with the number of members in a household. Concerning education, having secondary and tertiary education was positively related to WTP, suggesting that more educated respondents were willing to pay more. Regarding income, being in the high-income category ($\geq$ Zw\$3000) positively influenced WTP, meaning that wealthier individuals were more willing to pay for NHI. In addition, a positive perception of health insurance positively impacted WTP.

To obtain the mean WTP, Lopez-Feldman's 'nlcom' command was applied on the significant explanatory variables [46,48,49]. The mean WTP was found to be Zw\$72.13 per capita per month. Based on the market exchange rate that prevailed during the survey period [23], this translated to about US\$2.06 per person per month.

**Results of diagnostic tests.** In the WTJ and WTP analyses, the method of variance inflation factors (VIF) was used to test for multicollinearity, and a mean VIF of 1.73 (ranging between 1.04 and 3.98) was obtained. Thus, no severe multicollinearity among the regressors was present. Concerning the goodness-of-fit of the binary regression models, the results of the

**Table 9. Logit regression results of Willingness to Pay and the marginal effects.**

| Variable | Coefficient | Z statistic | P-value | Marginal effects |
|---|---|---|---|---|
| Age | -0.0041 (0.0193) | -0.21 | 0.831 | -0.0006 |
| Gender (ref: female) Male | -0.1097 (0.3432) | -0.03 | 0.975 | -0.0016 |
| Marital status (ref: unmarried) Married | -0.6015 (0.4236) | -1.42 | 0.156 | -0.0835 |
| Household size | -0.2549 (0.1047) | -2.44 | 0.015** | -0.0364 |
| Membership (ref: no) Yes | -0.1492 (0.3469) | -0.43 | 0.667 | -0.0211 |
| Awareness (ref: no) Yes | 0.0752 (0.6730) | 1.10 | 0.917 | 0.0101 |
| Perception (ref: no) Yes Not sure | 0.8529 (0.3578) 1.1955 (0.7709) | 2.38 1.55 | 0.017** 0.121 | 0.1271 0.1742 |
| Education (ref: primary) Secondary Tertiary | 1.4702 (0.5899) 2.1588 (0.8184) | 2.49 2.64 | 0.013** 0.008*** | 0.2258 0.3186 |
| Income (ref: < Zw$500) Zw$500 to < Zw$1000 Zw$1000 to < Zw$3000 ≥ Zw$3000 | -0.4308 (0.6086) 0.3002 (0.5918) 1.4346 (0.6578) | -0.71 0.50 2.18 | 0.749 0.615 0.029** | -0.0674 0.0455 0.1970 |
| Distance | -0.0871 (0.0836) | -1.05 | 0.294 | -0.0125 |
| Solidarity (ref: no) Yes | 0.0121 (0.6120) | 0.02 | 0.984 | 0.00173 |
| Difficulties (ref: no) Yes | 0.1835 (0.3707) | 0.50 | 0.620 | 0.0263 |
| Quality (ref: poor) Satisfactory Good Very good | 1.0529 (0.7092) 0.1463 (0.5637) - | 1.48 0.26 | 0.138 0.795 | 0.1386 0.0220 - |
| Bid price | -0.0525 (0.0066) | -7.98 | 0.000*** | -0.0075 |

(*Continued*)

**Table 9.** (Continued)

| Variable | Coefficient | Z statistic | P-value | Marginal effects |
|---|---|---|---|---|
| Constant | 3.4846 | | | |
| Log likelihood | -141.4866 | | | |
| Prob > Chi$^2$ | 0.0000 | | | |
| Pseudo R$^2$ | 0.3263 | | | |
| Observations | 321 | | | |

Standard errors are shown in parentheses.

Significance levels are shown by asterisks (

***p < 0.01

**p < 0.05).

**Table 10. Odds ratios for Willingness to Pay.**

| Variable | Odds ratio | Standard error | z-statistic | p-value |
|---|---|---|---|---|
| Age | 0.9959 | 0.0192 | -0.21 | 0.831 |
| Gender (ref: female) | | | | |
| Male | 0.9891 | 0.3395 | -0.03 | 0.975 |
| Marital status (ref: unmarried) | | | | |
| Married | 0.5480 | 0.2322 | -1.42 | 0.156 |
| Household size | 0.7750 | 0.0811 | -2.44 | 0.015** |
| Membership (ref: no) | | | | |
| Yes | 0.8614 | 0.2988 | -0.43 | 0.667 |
| Awareness (ref: no) | | | | |
| Yes | 1.0731 | 0.7222 | 0.10 | 0.917 |
| Perception (ref: no) | | | | |
| Yes | 2.3465 | 0.8397 | 2.38 | 0.017** |
| Not sure | 3.3051 | 2.5478 | 1.55 | 0.121 |
| Education (ref: primary) | | | | |
| Secondary | 4.3500 | 2.5661 | 2.49 | 0.013** |
| Tertiary | 8.6601 | 7.0871 | 2.64 | 0.008*** |
| Income (ref: < Zw$500) | | | | |
| Zw$500 to < Zw$1000 | 0.6500 | 0.3956 | -0.71 | 0.479 |
| Zw$1000 to < Zw$3000 | 1.3501 | 0.8069 | 0.50 | 0.615 |
| ≥ Zw$3000 | 4.1981 | 2.7616 | 2.18 | 0.029** |
| Distance | 0.9160 | 0.0766 | -1.05 | 0.294 |
| Solidarity (ref: no) | | | | |
| Yes | 1.0122 | 0.6194 | 0.02 | 0.984 |
| Difficulties (ref: no) | | | | |
| Yes | 1.2015 | 0.4453 | 0.50 | 0.620 |
| Quality (ref: poor) | | | | |
| Satisfactory | 2.8659 | 2.0326 | 1.48 | 0.138 |
| Good | 1.1575 | 0.6525 | 0.26 | 0.795 |
| Very good | - | - | - | - |
| Bid price1 | 0.9488 | 0.0063 | -7.98 | 0.000*** |

Constant: 32.6089

Log likelihood: -141.4866

Prob > Chi$^2$: 0.0000

Pseudo R$^2$: 0.3263

Observations: 321

Significance levels are shown by asterisks (

***p < 0.01

**p < 0.05).

**Table 11. Interval regression results of Willingness to Pay.**

| Variable | Coefficient | Standard error | Z statistic | P-value |
|---|---|---|---|---|
| Age | -0.3843 | 0.2537 | -1.51 | 0.130 |
| Gender (Male = 1) | 5.0711 | 4.5953 | 1.10 | 0.270 |
| Marital status (Married = 1) | -8.1367 | 5.4623 | -1.49 | 0.136 |
| Education (Secondary = 1) | 22.9739 | 7.7648 | 2.96 | 0.003*** |
| (Tertiary = 1) | 32.5736 | 10.5819 | 3.08 | 0.002*** |
| Income (Zw$500 to <Zw$1000 = 1) | -5.7906 | 8.0536 | -0.72 | 0.472 |
| (Zw$1000 to <Zw$3000 = 1) | 8.2288 | 8.0355 | 1.02 | 0.306 |
| (≥ Zw$3000 = 1) | 20.4634 | 8.6081 | 2.38 | 0.017** |
| Household size | -3.6273 | 1.3059 | -2.79 | 0.005*** |
| Distance | -1.4508 | 1.1409 | -1.27 | 0.204 |
| Membership (Yes = 1) | -3.1478 | 4.6442 | -0.68 | 0.498 |
| Awareness (Yes = 1) | -6.8224 | 7.3822 | -0.92 | 0.355 |
| Perception (Yes = 1) | 12.1028 | 4.6207 | 2.62 | 0.009*** |
| Solidarity (Yes = 1) | -2.9891 | 9.0218 | -0.33 | 0.740 |
| Difficulties (Yes = 1) | 2.2100 | 5.1284 | 0.43 | 0.667 |
| Quality (Satisfactory = 1) | 11.6242 | 8.3621 | 1.39 | 0.164 |
| (Good = 1) | 4.2882 | 7.7737 | 0.55 | 0.581 |
| (Very good = 1) | 0.4022 | 26.2288 | 0.02 | 0.988 |
| Constant | 81.7311 | 19.1040 | 4.28 | 0.000*** |
| Sigma | 30.6152 | 1.9093 | 16.03 | 0.000*** |

Wald $x^2$(18): 79.48
Prob > $x^2$: 0.0000
Log Likelihood: -347.8742
Observations: 323

Significance levels are shown by asterisks (

***p < 0.01

**p < 0.05).

Hosmer-Lemeshow test showed that each model was correctly specified; in all cases, the probability value was greater than the minimum threshold of 0.05.

## Discussion

### Willingness to join

Out of the 388 respondents who participated in the study, 325 (83.8%) expressed WTJ the proposed NHI scheme. The proportion of respondents that indicated a willingness to participate in health insurance studies conducted in other countries ranged between 69.5% and 92% [24,50–53]. Multivariate regression analysis showed that WTJ the proposed contributory NHI scheme among participants in selected urban informal sector clusters in Harare was influenced by health insurance awareness, membership to a resource-pooling scheme, solidarity with the sick, and experiencing difficulties paying for healthcare expenses in the past.

Concerning the health insurance variables (health insurance awareness and health insurance perception), the study established positive relationships with WTJ. Respondents who had awareness of private health insurance schemes in Zimbabwe were more willing to join. An investigation of the determinants of WTJ some Community Based Health Insurance (CBHI) schemes in Cameroon established that lack of awareness of the existence of such schemes among the target population significantly explained low participation [54]. Despite the government of Cameroon establishing CBHI schemes in each health district in 2006, more than 5 years later, Noubiap et al. [54] found that only 25.6% of the sampled respondents knew about the presence of those schemes in their district. Such was also the case in Nigeria, where an NHI scheme was introduced in 2000, but a study conducted 17 years later found that 41.3% of the respondents had never heard about the scheme [55]. A common factor for the low levels of awareness cited by the two studies was low educational attainment–generally, those without primary education were unaware. Nevertheless, it is not the mere awareness of health insurance schemes that matters–adequate awareness (detailed knowledge) about the services offered and the benefits of being a member matter [56]. Good knowledge/ awareness of CBHI positively influenced WTJ in Ethiopia [57].

The study also found that respondents who were current members of a resource-pooling scheme such as private health insurance, funeral fund or savings and lending club were more willing to join. A Kenyan study on willingness to participate in Social Health Insurance (SHI) found that being a member of a Savings and Credit Cooperative Organisation (SACCO) or a community savings group significantly positively influenced WTJ [58]. Membership in resource-pooling schemes fosters a better understanding of risk-sharing and, in turn, influences the probability of someone participating in public health insurance initiatives.

Willingness to help others on health matters (solidarity with the sick) positively influenced WTJ. Respondents who answered "yes" to contributing towards healthcare costs for someone sicker than themselves were more willing to join the proposed NHI scheme. A study by James and Savedoff [59] established that Zimbabweans generally empathise with the sick and poor, and display the spirit of solidarity in times of sickness. A high level of solidarity in society creates fertile ground for the successful implementation of NHI [60]. Participation in community groups when there is sickness or death in the community positively impacted WTJ CBHI in Ethiopia [61].

Yet another significant determinant of WTJ was the household having experienced difficulties paying for healthcare in the past 12 months. Respondents whose households had experienced challenges meeting treatment costs were more willing to join. The higher likelihood of WTJ a health insurance scheme is understandable as such respondents are willing to join to be able to easily cope with future healthcare expenses [45,62].

## Willingness to pay

Of the 325 respondents who were willing to join, 208 (64%) answered "yes" to paying the first bid that was offered. The researcher randomly drew the initial bid from four bids (Zw$25, Zw$50, Zw$75 and Zw$100), and this was done to alleviate the problem of bid anchoring where respondents tend to state a maximum WTP amount not very far from the one initially offered by the investigator. Results from the DBDC with follow-up elicitation showed that the mean WTP of the sampled informal sector participants was Zw$72.13 per person per month. When converted to US dollars at the market exchange rate that prevailed at the time of the research [23], the average WTP was US$2.06 per capita per month. In comparison to studies that elicited WTP per person per month in other low- and middle-income countries (LMICs), the mean WTP obtained in this study (US$2.06) is not significantly different from that obtained in

Cameroon (US$2.15) [63], and in Nigeria (US$1.68) [53]. However, the figure was lower than that found in Sierra Leone (US$3.60) and Namibia (US$6.60) [45,64], and much lower than the one from a study in St Vincent and the Grenadines (US$28.23) [50]. Of course, the segment of the population surveyed as well as the income status of a country (high, middle, or low) is an important factor in explaining such variations in mean WTP. As such, it is more appropriate to compare the mean WTP of this study to that of studies that had wide coverage of informal sector participants from either rural or urban settings, or both [45,53,63]. Moreover, the studies were conducted in low-income countries [45] or lower-middle-income economies [53,63] and thus their results can be easily compared to those of Zimbabwe (a lower-middle-income nation).

Considering the difficult economic situation prevailing in Zimbabwe, especially after the re-introduction of a quasi-local currency in 2016, the average WTP of Zw$72.13 per person per month could be steep for the average informal sector participant in the sampled clusters. While the premium corresponded to at most 2.4% of the income of informal sector workers in the highest-income category ($\geq$ Zw$3000), it turned out to be at least 14.43% of the income of those falling in the lowest category ($<$ Zw$500) who constituted 8.8% of the study sample. Nevertheless, when expressed as a percentage of the 2019 per capita GDP for Zimbabwe [65], the mean WTP of $2.06 per person per month corresponds to approximately 1.69% of per capita GDP, a figure which is higher than that estimated by Nosratnejad et al. [31] for LMICs (1.18% of GDP per capita for individual member contributions). Given that funds that can be raised through individual or household insurance premiums are limited, government financing remains crucial even after the introduction of NHI. Other literature suggests government healthcare spending of at least 5% of GDP for moving closer to UHC [66]. In Ethiopia, the government pays a general subsidy of 25% on premiums of all CBHI members in that country [67].

Borrowing from related studies, levying a flat premium on all informal sector participants may not be appropriate given the people's diverse socioeconomic backgrounds; instead, setting premiums on a sliding scale seems a better alternative [45,53,68]. As the researchers observe, the main challenge associated with such an option is to determine the incomes of informal sector participants. Perhaps, one way to ameliorate this challenge could be to allow for "health insurance package differentiation" whereby variations in packages are made so that members can choose their preferred package depending on their ability to pay. High-income earners might be attracted by unique package features such as additional benefits or seeking treatment at certain facilities or in special wards.

With respect to the determinants of WTP for the proposed NHI scheme, the logit model (SBDC elicitation) and the interval data model yielded similar results. The results show that the significant predictors of WTP for NHI among participants from the selected urban informal sector clusters are household size, education, income, and health insurance perception. On household size, a negative relationship with WTP existed–the more the number of members in a household, the lower the WTP. While most studies obtained a positive relationship between household size and WTP [31,52,68,69], the negative association observed in the current study is plausible. The sign of the coefficient is largely determined by the way the valuation scenario is framed [51]. If each household member is required to pay a premium, respondents are less willing to pay, and the reverse is true if the premium is paid at the household level. In their study on WTP for health insurance in Saudi Arabia, Al-Hanawi et al. [51] got a negative significant relationship between household size and WTP.

Education of the respondent was also a significant predictor of WTP. Respondents who completed secondary and tertiary education were more willing to pay. Some previous studies found a positive relationship between education and WTP [31,45,52,68,69] while others

obtained a negative relationship [24,53]. The positive and significant coefficients for secondary and tertiary education in this study may imply that more educated informal sector participants are more cognisant of the need to financially contribute to the resuscitation of the country's public healthcare delivery system.

The study also established that being in the highest-income category ($\geq$ Zw\$3000 per month) was positively and significantly related to WTP. Although after a certain income threshold, individuals may become less willing to pay for health insurance [20], most empirical studies have found a positive relationship between the two [21,24,31,52,53,69]. Thus, the results of this study concur with those of past research in the field. However, especially for informal sector participants, periodic fluctuations in income are an important issue for consideration. To account for such income fluctuations, it might be necessary to come up with flexible premium payment arrangements [53] based on the inputs of informal sector participants. This may help in retaining membership and at the same time encouraging new enrolees to join.

The multivariate regression analyses also established a positive, significant relationship between health insurance perception and WTP–respondents who had a positive perception of the value/ benefit of health insurance were more willing to pay. The results are supported by related literature. It is argued that low awareness of the value of insurance negatively influences WTP [31]. Awareness of existing CBHI schemes was found to be a significant predictor of WTP for health insurance in Ethiopia [70]. Nevertheless, of concern in the present study was that a significant proportion of those who were active members of a private health insurance scheme (33.3%) had a negative perception of the value of health insurance. The unstable macroeconomic environment in Zimbabwe saw commodity prices being frequently adjusted upwards, and this was also the case with medical drugs, health insurance premiums, and co-payments for health services. Private health insurance holders were being asked to make additional payments, not in the form of the usual co-payments based on the health insurance contract, but rather as shortfalls arising from escalating medical costs [71,72]. In this regard, macroeconomic stability is a fundamental pillar for the acceptance of NHI among the urban informal sector workers.

The logit regression for WTP established a negative and significant relationship between WTP and the initial bid price. This result conforms to economic theory; according to the law of demand, an inverse relationship exists between price and quantity demanded. The negative and significant result means that the probability of answering "yes" to the WTP question was significantly influenced by the bid price.

## Limitations of the study

The study was conducted in an environment where the concept of NHI was not well-known among respondents. To account for this knowledge gap, the researcher briefly explained the concept of a contributory NHI scheme as respondents were taken through the participant information sheet. Moreover, as part of the questionnaire, an NHI scenario was presented to enhance respondents' understanding of the concept.

By its nature, the contingent valuation technique deals with intended rather than actual behaviour. Consequently, the method is prone to hypothetical bias as respondents may not take the valuation exercise as culminating in the provision of the promised public good. In this study, the problem was mitigated by using a coercive payment vehicle (insurance premiums) that reduces the probability of free riding. Moreover, the premiums offered to respondents in the DBDC elicitation were realistic since they were informed by contributions that were being made to private health insurance providers in Zimbabwe. Also, during the survey the

researcher highlighted having obtained permission to undertake the research from the MoHCC.

Health decisions are made at the household level, and normally the household head completes the questionnaire on behalf of the family. However, in the current study, it was not possible to collect data exclusively from household heads since data collection happened at the workplace rather than at home. It was not possible to apply probability sampling at homes because such a sampling frame was unavailable; the Harare City Council's Human Capital Department could only provide statistics on the number of stalls by informal sector cluster. Nevertheless, this information enabled the use of probability sampling to select respondents at their workplaces. The use of probability sampling is in line with the best practice for stated preference studies. Moreover, no children under 18 years old nor the elderly (over 64 years old) responded to the questionnaire. Thus, although some respondents were not household heads, the researcher targeted individuals capable of making decisions on behalf of households.

## Conclusions

The study explored the WTJ and WTP for a contributory NHI scheme among Zimbabwe's urban informal sector participants from selected urban informal sector clusters of Harare. The contingent valuation technique was used to elicit WTP. Results of the study showed that out of the 388 respondents, 325 (83.8%) were willing to join the proposed NHI scheme. This result, to an extent, indicates the importance that informal sector participants from the targeted clusters attach to pre-payment arrangements for healthcare. WTJ was influenced by the following factors: health insurance awareness, health insurance perception, membership to a resource-pooling scheme, solidarity with the sick, and household having experienced difficulties paying for healthcare during the past 12 months. With respect to WTP, on average, the sampled informal sector participants were willing to pay Zw$72.13 per person per month, which was equivalent to about US$2.06 per capita per month at the time of the survey. The key determinants of WTP were household size, respondent's level of education, income, and health insurance perception. Since the majority of respondents were willing to join and pay for a contributory NHI scheme, it follows that there is potential to introduce such a scheme for the urban informal sector workers in the sampled clusters.

However, some issues require careful consideration before such a scheme can be rolled out. Communities need to be educated on the concept of risk pooling and the benefits of being members of the NHI scheme. There is a need for government to sensitize the informal sector communities about the scheme and ensure that they get some fair understanding of its modalities–mere awareness of the scheme launch is not enough. Also, household size and income are factors that require special attention when deciding on the premiums for the scheme as affordability is an important determinant of the size of the risk pool. Moreover, given that price instability hurts financial products such as health insurance, there is a need for ensuring macroeconomic stability (stable exchange rate and price level). Not only will this preserve the values insured and boost confidence among potential enrolees; the capacity of the informal sector participants to contribute to the risk pool is also enhanced.

## Supporting information

**S1 File. Survey questionnaire.**
(DOC)

**S2 File. Informed consent.**
(DOC)

**S3 File. Survey data.**
(XLSX)

## Acknowledgments

The authors are grateful to Zimbabwe's MoHCC and the Harare City Council for granting permission to conduct this study and collect data from urban informal sector participants in Harare. The authors are also indebted to the people who work in the following informal sector clusters of Harare: Glenview furniture complex, Mupedzanhamo flea market, Harare home industries, and Mbare markets. Without their valuable input, this study would not exist.

## Author Contributions

**Conceptualization:** Tamisai Chipunza.

**Data curation:** Tamisai Chipunza.

**Formal analysis:** Tamisai Chipunza.

**Investigation:** Tamisai Chipunza.

**Methodology:** Tamisai Chipunza.

**Supervision:** Senia Nhamo.

**Writing – original draft:** Tamisai Chipunza.

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
