## [Decision Letter · Decision Letter 0]

7 Feb 2023

PONE-D-22-35610Potential demand for National Health Insurance in Zimbabwe: evidence from selected informal sector clusters in HararePLOS ONE

Dear Dr. Chipunza,

Thank you for submitting your manuscript to PLOS ONE. After careful consideration, we feel that it has merit but does not fully meet PLOS ONE’s publication criteria as it currently stands. Therefore, we invite you to submit a revised version of the manuscript that addresses the points raised during the review process. Your manuscript has been assessed by two reviewers whose reports can be found below and in the attached documents. As you will see from the comments, the reviewers point out methodological issues that should be addressed before your manuscript is suitable for publication. Please follow the PLOS ONE manuscript format.

Please submit your revised manuscript by Mar 24 2023 11:59PM. If you will need more time than this to complete your revisions, please reply to this message or contact the journal office at plosone@plos.org. Please include the following items when submitting your revised manuscript:A rebuttal letter that responds to each point raised by the academic editor and reviewer(s). You should upload this letter as a separate file labeled 'Response to Reviewers'.A marked-up copy of your manuscript that highlights changes made to the original version. You should upload this as a separate file labeled 'Revised Manuscript with Track Changes'.An unmarked version of your revised paper without tracked changes. You should upload this as a separate file labeled 'Manuscript'.

We look forward to receiving your revised manuscript.

Kind regards,

Rornald Muhumuza Kananura

Academic Editor

PLOS ONE

Journal Requirements:

2. Please ensure that you have specified (1) whether consent was informed and (2) what type you obtained (for instance, written or verbal, and if verbal, how it was documented and witnessed). If your study included minors, state whether you obtained consent from parents or guardians. If the need for consent was waived by the ethics committee, please include this information.

Reviewers' comments:

Reviewer's Responses to Questions

**Comments to the Author**

1. Is the manuscript technically sound, and do the data support the conclusions?

Reviewer #1: Yes

Reviewer #2: Yes

2. Has the statistical analysis been performed appropriately and rigorously? 

Reviewer #1: Yes

Reviewer #2: No

3. Have the authors made all data underlying the findings in their manuscript fully available?

Reviewer #1: No

Reviewer #2: Yes

4. Is the manuscript presented in an intelligible fashion and written in standard English?

Reviewer #1: Yes

Reviewer #2: Yes

5. Review Comments to the Author

Reviewer #1: The paper Potential Demand for National Health Insurance in Zimbabwe is well written, clear and the methods sound. The discussion is also well argued. Nevertheless, I have a few concerns:

1. Sample size: The informal economy in Zimbabwe is estimated to be 40% of its GDP. Between 80 to 90% of the Zimbabweans are engaged in informal economic activities. Therefore, the approximately 400 respondents calculated on a cluster of informal activity of about 4,000 seems too restrictive to me, or probably not representative of the informal sector in the country. Moreover, they all seem to be leaving in an urban setting (as rural/urban is not a control).

2. Results:

2.1. Willingness to join: why was price not included? an explanation would have helped understand. I fear that some of the regressors (perception of HI) are highly endogeneous. The unobserved variables that explain the perception may underlie the willingness to join (and to participate).

2.3. Willingness to pay: As the household size is a proxy of the price paid, it is not surprising that each additional member lowers the probability of WTP for health insurance. I wonder if this should have been interacted with the initial bid to understand better the effect of household size separately from the effect of the price. One would expect that if the effects can be isolated, the number of members in the household would have a positive effect.

3. Originality: The contribution of this paper is of interest as another willingness to participate and pay for health insurance schemes in a LMICs but based on a very small sample of one particular country. It might be of interest to the policy makers in that country of that country but it is difficult to see how at this point it is of general interest.

Minor points:

4. There is no need to present both probit and logit. One of the specifications could have been chosen based on the distribution of the dependent variable. It does not add much to have both.

5. Minor: Tables 2 to 6 were not available to the referee. The univariate and bivariate descriptive analysis is usually described as descriptive analysis.

Reviewer #2: GENERAL COMMENTS

Health decisions are mainly decisions made at the household level especially for poor and informal sector households. So why did you use individuals as against households as your economic agent. Please explain briefly even in two sentences your choice of economic agents.

You need to define every acronym you used

Please go over your methodology especially on WTP and Marginal WTP for the NHI.

In your methodology, you did not explain any model and how you came to the conclusion of using the regression methods you chose.

Generally, you need to explain with reasons why your choice of a particular method over the others.

Also ensure that the values in the results are what’s being explained in your analysis.

SEE ATTACHED DOCUMENT FOR ADDITIONAL COMMENTS

6. PLOS authors have the option to publish the peer review history of their article (what does this mean?). If published, this will include your full peer review and any attached files.

Reviewer #1: No

Reviewer #2: No

---

## [Author Response · Author response to Decision Letter 0]

2 Mar 2023

COMMENTS ON MANUSCRIPT TITLED - POTENTIAL DEMAND FOR HEALTH INSURANCE IN ZIMBABWE: EVIDENCE FROM INFORMAL SECTOR CLUSTERS IN HARARE

GENERAL COMMENTS

Health decisions are mainly decisions made at the household level especially for poor and informal sector households. So why did you use individuals as against households as your economic agent. Please explain briefly even in two sentences your choice of economic agents.

Response 

It’s true that health decisions are made at the household level, and normally the household head responds to the questionnaire on behalf of the family. However, in the current study it was not possible to collect data exclusively from household heads since data collection happened at the workplaces rather than at homes. It was not possible to apply probability sampling at homes because such sampling frame was unavailable; the Harare City Council’s Human Capital Department could only provide statistics on the number of stalls by informal sector cluster and this information enabled the use of probability sampling to select respondents at their workplace. The use of probability sampling is in line with the best practice for stated preference studies. Moreover, no children under 18 years old nor the elderly (over 64 years) responded to the questionnaire. Nevertheless, failure to interview strictly household heads only has been documented as one of the study limitations (see the revised manuscript).

You need to define every acronym you used

Response

All acronyms have now been defined – see the revised manuscript.

Please go over your methodology especially on WTP and Marginal WTP for the NHI.

Response

Models for WTJ and WTP have been specified and described (see the revised manuscript)

In your methodology, you did not explain any model and how you came to the conclusion of using the regression methods you chose.

Response

The logit model for Willingness to Join and Pay the first bid, and the interval data model for maximum WTP have now been presented and described in the revised manuscript.

Generally, you need to explain with reasons why your choice of a particular method over the others.

Response

The use of certain regression methods over others has now been explained (see the revised manuscript).

Also ensure that the values in the results are what’s being explained in your analysis. 

Response

All interpretations have been crosschecked to ensure that they are in line with the values reported in the results.

SPECIFIC COMMENTS

Abstract – In your methods, you failed to show the exact category of Informal Sectors you included in the study. It will be good to know those categories.

Response

The names of the informal sector clusters studied have also been included in the abstract (see the revised manuscript). Also, the word “urban” has been included in the manuscript title and whenever the informal sector clusters are mentioned so that readers can easily recognize that Harare is an urban area. 

Study Design – As stated above, there is need to know the informal sectors we’re talking of here. So please list/name them for better understanding.

Response

The names of the informal sector clusters had already been stated under the Study Design section of the original manuscript (see the original and revised manuscripts). 

Eliciting Willingness to Join and Pay

The word contingent in first sentence is the wrong word in that sentence. Rephrase the sentence with the right word to show exactly what you want to explain.

Response

The word “contingent” has been replaced to add clarity to the meaning of the statement (see the revised manuscript).

The DBDC with follow up is not properly explained. So, add few sentences to make it clearer for the reader. Note, it is the basis for the estimation techniques and results used in this work.

Response

Some sentences have been added to make the DBDC with follow-up method clearer to the reader (see the revised manuscript).

Line 6 in paragraph 3 should be rephrased to read thus: If the respondent indicated WTP, a higher bid was offered as the starting premium.

Response

A higher bid was not offered as the “starting premium”; instead, the higher bid was offered as a “follow up to accepting the initial bid” – this is highlighted in section D of the questionnaire. In the revised manuscript, the researcher has provided further details to clarify this issue. 

In your methodology, you did not state reasons why you choose the DBDC with follow up above the other CV methods. 

Response

The relative merits of the DBDC with follow-up method in comparison to other elicitation formats have now been highlighted (see the revised manuscript).

Willingness to Join and Pay

What is the rationale for using both regression methods – Logit and Probit Models. You need to explain why the decision for using both models. I’ll prefer you use one over the other but with a reason.

Response

There was no big reason for estimating both the logit and probit models. In the revised manuscript, only the logit model was estimated as it allowed the researcher to report and interpret the odds ratios as well; the goodness of fit of the logit and probit models was generally the same.

Results

It is confusing having some tables and the main tables that should be shown in the work are not there. Instead, you have directed us the readers to insert S3, S4, S5 etc. to see the tables but others where shown in the text. Be consistent in that and preferably show the main tables and the others send them to an appendix.

Response

The researcher placed some tables as “inserts” to avoid making the manuscript relatively long. In the revised manuscript, most tables are now shown in the text – only those for the odds ratios remain in the appendix.

Univariate Analysis

Those individuals that did not want to join the health insurance scheme, did they give any reason? If yes, it will be nice to know those reasons.

Response

The researcher did not make follow-ups to get the reasons for not willing to join.

Willingness to Pay for NHI

Respondents who were WTJ were presented with the initial bid randomly drawn from 4 bids where the respondents tend to state their maximum WTP amount. The amount can take the form of any of those 4 bids; hence a Multinomial Logit Model can be appropriate here since it can handle more than two choices.

Response

The multinomial logit was not applicable because the respondent was not given the opportunity to choose among the 4 initial bids; instead, only one initial bid was offered to the respondent depending on the questionnaire that was randomly drawn by the researcher. Thus, the first WTP response in the “one and only one” questionnaire presented to the respondent was binary (yes or no answer). In the logit model for WTP, the initial bid price was included as one of the control variables. As highlighted in the original and the updated manuscripts, the researcher prepared 4 sets of questionnaires which only differed on the bids offered (section D) and this was done to minimize anchoring bias. For further clarification, the 4 versions of the questionnaire (which only differ on section D) are embedded as a single document here and the same has been attached as supporting information:

Note also that when the DBDC with follow up model is used, we get 4 possible outcomes, with probabilities thus:

Pr. (Yes, Yes)

Pr. (Yes, No)

Pr. (No, Yes)

Pr. (No, No)

You model this and use a loglikelihood function. Thus, an OLS is not the appropriate method here also. There are other estimation methods to use, check the literature properly.

Response

It’s true that with the DBDC elicitation there are 4 possible outcomes, and estimation should be done via maximum likelihood since the regressand will be bounded between two points. However, it is argued that if respondents also state their maximum Willingness to Pay (WTP), the regressand will be a continuous variable with a single point estimate such that the OLS technique can be applied (https://doi.org/10.3390/su132111711). Studies that elicited WTP using the DBDC method and estimated WTP via OLS include https://doi.org/10.3390/su132111711; https://doi.org/10.1186/s12939-020-01343-9; and https://doi.org/10.3389/fpubh.2021.654362.

In the revised manuscript, a maximum likelihood approach (interval regression) was applied for the DBDC WTP model, and the results validated those of the OLS technique adopted earlier.

Thank you

---

## [Decision Letter · Decision Letter 1]

28 Mar 2023

PONE-D-22-35610R1Potential demand for National Health Insurance in Zimbabwe: evidence from selected urban informal sector clusters in HararePLOS ONE

Dear Dr. Chipunza,

Thank you for submitting your manuscript to PLOS ONE. After careful consideration, we feel that it has merit but does not fully meet PLOS ONE’s publication criteria as it currently stands. Therefore, we invite you to submit a revised version of the manuscript that addresses the points raised during the review process.

 Please address the following minor comments before it is forwarded for publication. -Could you include the point of data collection (household or workplaces or both??) in the abstract and under methods. In an informal sector, a workplace can also be a place of residence. - also, let the reader know how the sampling was done (in the abstract). You indicate systematic sampling, but you don't tell how systematic it was and was it uniform across all clusters? Also, how were the clusters selected? purposively or randomly? and what was the sampling frame and how did you allocate the sample size? if it was purposive, please give a reason. - I am thinking it would be nice to include some socio-economic (income, occupation, education) include age of your participants in the abstract. - Please make all the variables in the table clear. for instance, Hhsize (I am supposing household size). - It would have been nice to generate the marginal probability graphs for some of the continuous variables (household size, age, distance) and with other variables with at least three categories (income, education level). -Conclusion (both abstract and main text) and recommendations should be based on your finding. Kindly tailor this to the study findings -under background section, I missed statistics on OOP in Zambabwe and the overall health financing system in the country. 

We look forward to receiving your revised manuscript.

Kind regards,

Rornald Muhumuza Kananura

Academic Editor

PLOS ONE

Journal Requirements:

Reviewers' comments:

Reviewer's Responses to Questions

**Comments to the Author**

1. If the authors have adequately addressed your comments raised in a previous round of review and you feel that this manuscript is now acceptable for publication, you may indicate that here to bypass the “Comments to the Author” section, enter your conflict of interest statement in the “Confidential to Editor” section, and submit your "Accept" recommendation.

Reviewer #2: All comments have been addressed

2. Is the manuscript technically sound, and do the data support the conclusions?

Reviewer #2: Yes

3. Has the statistical analysis been performed appropriately and rigorously? 

Reviewer #2: Yes

4. Have the authors made all data underlying the findings in their manuscript fully available?

Reviewer #2: Yes

5. Is the manuscript presented in an intelligible fashion and written in standard English?

Reviewer #2: Yes

6. Review Comments to the Author

Reviewer #2: THE AUTHORS HAVE ADDRESSED MY PREVIOUS COMMENTS ADEQUATELY AND IN A COHERENT MANNER. HAVING GONE THROUGH THIS ROUND OF THE MANUSCRIPT, I CAN RECOMMEND NOW FOR IT TO BE PUBLISHED.

7. PLOS authors have the option to publish the peer review history of their article (what does this mean?). If published, this will include your full peer review and any attached files.

Reviewer #2: **Yes: **Joseph Kamara Ph.D.

---

## [Author Response · Author response to Decision Letter 1]

23 Apr 2023

PONE-D-22-35610R1

Potential demand for National Health Insurance in Zimbabwe: evidence from selected urban informal sector clusters in Harare

PLOS ONE

Dear Dr. Chipunza,

Thank you for submitting your manuscript to PLOS ONE. After careful consideration, we feel that it has merit but does not fully meet PLOS ONE’s publication criteria as it currently stands. Therefore, we invite you to submit a revised version of the manuscript that addresses the points raised during the review process.

Please address the following minor comments before it is forwarded for publication.

-Could you include the point of data collection (household or workplaces or both??) in the abstract and under methods. In an informal sector, a workplace can also be a place of residence.

Response

The point of data collection (that is, a workplace) has been specified in the abstract and under methods. (See the revised manuscript). 

- also, let the reader know how the sampling was done (in the abstract). You indicate systematic sampling, but you don't tell how systematic it was and was it uniform across all clusters? Also, how were the clusters selected? purposively or randomly? and what was the sampling frame and how did you allocate the sample size? if it was purposive, please give a reason.

Response

The use of purposive sampling in the first stage, the nature of the systematic sampling applied thereafter, and the proportional allocation of the sample based on cluster size have been described in both the abstract and the methods section. Justification for using purposive sampling in the first stage has also been provided in the methods section. (See the revised manuscript).

- I am thinking it would be nice to include some socio-economic (income, occupation, education) include age of your participants in the abstract.

Response

Information on the occupation, education, income, and age of respondents has been included in the abstract (see the revised manuscript).

- Please make all the variables in the table clear. for instance, Hhsize (I am supposing household size).

Response

Full words or appropriate phrases (rather than abbreviations) have been used to describe variables in all the tables. (See the revised manuscript).

- It would have been nice to generate the marginal probability graphs for some of the continuous variables (household size, age, distance) and with other variables with at least three categories (income, education level).

Response

Graphs for some categorical variables (income, education level, employment status), and the continuous variables (household size, age, distance) have been generated (see Figs 1A-1C and Figs 2A-2C of the revised manuscript). 

-Conclusion (both abstract and main text) and recommendations should be based on your finding. Kindly tailor this to the study findings

Response

Clarity has been added to ensure that the conclusions and recommendations relate to the findings from informal sector clusters that were studied.

-under background section, I missed statistics on OOP in Zambabwe and the overall health financing system in the country.

Response

Further information on the existing healthcare financing model in Zimbabwe has been provided in the introduction section of the revised manuscript.

Thank you

---

## [Editor Report · Decision Letter 2]

16 May 2023

Potential demand for National Health Insurance in Zimbabwe: evidence from selected urban informal sector clusters in Harare

PONE-D-22-35610R2

Dear Dr. Chipunza,

We’re pleased to inform you that your manuscript has been judged scientifically suitable for publication and will be formally accepted for publication once it meets all outstanding technical requirements.

Kind regards,

Rornald Muhumuza Kananura, PhD

Academic Editor

PLOS ONE
---

## [Editor Report · Acceptance letter]

18 May 2023

PONE-D-22-35610R2 

Potential demand for National Health Insurance in Zimbabwe: evidence from selected urban informal sector clusters in Harare 

Dear Dr. Chipunza:

I'm pleased to inform you that your manuscript has been deemed suitable for publication in PLOS ONE. Congratulations! Your manuscript is now with our production department. 

Kind regards, 

on behalf of

Dr. Rornald Muhumuza Kananura 

Academic Editor

PLOS ONE